# Loneliness among Homeless Individuals during the First Wave of the COVID-19 Pandemic

**DOI:** 10.3390/ijerph18063035

**Published:** 2021-03-16

**Authors:** Franziska Bertram, Fabian Heinrich, Daniela Fröb, Birgit Wulff, Benjamin Ondruschka, Klaus Püschel, Hans-Helmut König, André Hajek

**Affiliations:** 1Institute of Legal Medicine, University Medical Center Hamburg-Eppendorf, 22529 Hamburg, Germany; fa.heinrich@uke.de (F.H.); Daniela.froeb@uke.de (D.F.); Birgit.wulff@uke.de (B.W.); b.ondruschka@uke.de (B.O.); pueschel@uke.de (K.P.); 2Department of Health Economics and Health Services Research, University Medical Center Hamburg-Eppendorf, 22529 Hamburg, Germany; h.koenig@uke.de (H.-H.K.); a.hajek@uke.de (A.H.)

**Keywords:** homeless, UCLA-3, loneliness, Sars-CoV-2, COVID-19, sleeping rough, social isolation, social exclusion, coronavirus disease

## Abstract

The feeling of loneliness is a major public health concern associated with multiple somatic and psychiatric illnesses. Studies have shown increasing incidence of loneliness in the general population during the first wave of the COVID-19 pandemic. Homeless individuals are a particularly vulnerable group; however, little is known about loneliness among homeless individuals. We therefore aimed to examine the prevalence of loneliness among homeless individuals during the pandemic. Furthermore, we estimated the association between loneliness and sociodemographic and lifestyle factors, as well as the self-perceived risk of contracting COVID-19. Data from the Hamburg survey of homeless individuals were used, including 151 homeless individuals that were recruited in spring of 2020. Loneliness was measured by the 3- item version of the UCLA-3 Loneliness Scale. To summarize, 48.5% of the participants experienced loneliness. Multiple linear regressions showed increased loneliness to be associated with male gender (β = 1.07, *p* = 0.01), being single (β = 1.33, *p* = 0.00), originating from Germany (β = 1.48, *p* = 0.00), high frequency of sharing a sleeping space with more than three people (β = 0.42, *p* = 0.02) and a higher self-perceived risk of contracting COVID-19 (β = 0.41, *p* = 0.02). On the contrary, there was no association of loneliness with age, educational level, chronic alcohol consumption or frequently sharing a sleeping space. In conclusion, the magnitude of loneliness among homeless individuals during the pandemic was highlighted. Description of factors determining loneliness may help to identify homeless individuals at risk.

## 1. Introduction

The feeling of loneliness describes a discrepancy between actual and desired social relationships and is considered a major public health problem [1,2]. Loneliness is associated with multiple somatic and psychiatric illnesses, as well as premature death [3]. Meta-analysis found elevated relative risk for hypertension, cardiovascular disease and post myocardial infarction mortality in people experiencing loneliness. Furthermore, loneliness is associated with personality disorders, psychosis, depression and suicide [4]. The feeling of loneliness also impairs cognitive performance, increases the risk of cognitive decline and reduces executive control. Further highlighting the devastating consequences of loneliness, experimental evidence revealed that the feeling of loneliness also increases perceived stress, fear of negative evaluation, anxiety and anger by reducing optimism and a person’s self-esteem [1]. Cross-sectional studies before the pandemic show that in 2012, 10.5% of the German general population suffered from loneliness [5], a prevalence which increased after the implementation of social distancing policies during the COVID-19 pandemic [6].

Even before the pandemic, data from Spain, Australia and Canada showed that homeless individuals have a particularly high risk of experiencing social isolation and loneliness [7,8,9], showing that up to 39.6% of the homeless population feel lonely [10]. Hence, a particular high risk of loneliness combined with social distancing policies might explicitly expose homeless individuals to the physiologic and psychologic consequences of loneliness. According to the federal state law of the City of Hamburg, a homeless individual is defined as someone who does not have stable housing and has been sleeping rough, or in emergency shelters, for most days within the last month. Individual reasons for homelessness are diverse but frequently include poverty, family problems, substance misuse and the availability of low-cost housing [11]. Compared with the general population, homeless individuals have higher rates of premature death, increased prevalence of infectious and chronic diseases, as well as a high prevalence of mental disorders [11]. In 2018, an estimated 41,000 homeless individuals were living in Germany [12]; 6500 were in the federal state of Hamburg, the second largest city in Germany [13]. In March of 2020, the first wave of the COVID-19 pandemic hit Germany, leading to the implementation of social distancing policies and recommendations to protect the general public from infection with SARS-CoV-2, commonly known as COVID-19 [14]. Consequently, many overnight shelters, soup kitchens and medical practices closed to protect working staff and homeless individuals from contracting SARS-CoV-2 infections. Those changes in sheltering and the lack of opportunities to beg for food and money in public places led to a marked change of living conditions for homeless individuals [15]. 

During the COVID-19 pandemic, providing care for homeless individuals is of high importance, particularly because their living conditions make it difficult for homeless individuals to follow the recommendations for personal hygiene, which could facilitate virus transmission. Furthermore, homeless individuals can potentially become “super spreaders” due to their increased mobility [16]. Additionally, homeless individuals are particularly vulnerable to suffering from a more severe course of COVID-19 due to multiple pre-existing somatic and mental illnesses [11,17]. Evaluating the feeling of loneliness in people experiencing homelessness is highly important, as homeless individuals that are lonely are less likely to get in contact with counselors and social workers [18], or to visit physicians in the case of illnesses. In fact, establishing contact between homeless individuals and professionals is important to improve education about virus transmission, provide facilities and tools for personal hygiene and, in the case of SARS-CoV-2 infection, help with isolation and adequate treatment [17]. 

To this end, we aimed to investigate the feeling of loneliness among homeless individuals in Hamburg during the first wave of the COVID-19 pandemic. Furthermore, we determined sociodemographic and lifestyle factors associated with an elevated risk of loneliness, opening up opportunities to identify homeless individuals who could benefit from targeted interventions [19].

## 2. Materials and Methods

### 2.1. Sample

Data were collected in the Hamburg survey of homeless individuals. This survey was initialized to evaluate the psychologic and somatic health consequences of the COVID-19 pandemic on homeless individuals in Hamburg. Data collection was conducted from 25 May to 3 June 2020. In total, 151 people were recruited in shelters, lodging houses and medical practices offering specialized care for homeless individuals. An incentive of 5€ per 30 min was offered to all participants. The response rate was 98%. All participants reported experiencing homelessness for at least one month. Due to missing values, the analytical sample was composed of 130 individuals. Data on demographics, psychiatric and somatic illnesses were collected. All participants were interviewed in a separate room. In addition, a blood withdrawal was conducted to determine biological markers for chronic alcohol consumption. Where possible, participants were asked to fill out their questionnaires independently. Most individuals were interviewed face-to-face to help with problems they may have had reading or understanding the questions. The study was approved by the Ethics Committee of the Hamburg Chamber of Physicians (No.: PV7333), complying with the Declaration of Helsinki. The datasets analyzed in the current study are not publicly available due to ethical restrictions involving patient data but may be provided by the corresponding author upon reasonable request.

### 2.2. Dependent Variables

Loneliness was measured using the UCLA Loneliness Scale Version 3 (UCLA-3), a three-item scale for loneliness proposed by Hawkley and colleagues that was developed for use in large surveys. UCLA-3 is a commonly used questionnaire with high reliability and validity, consisting of three items: (1) “How often do you feel that you lack companionship?”; (2) “How often do you feel left out?”; (3) “How often do you feel isolated from others?”. Each question can be answered on a three-point Likert scale: 1 = “Hardly Ever”, 2 = “Some of the Time”, 3 = “Often” [20]. For analysis, a sum score ranging from 3 to 9 points was calculated, higher values indicate greater loneliness. Data from the UCLA-3 can also be interpreted categorically, defining individuals with a sum score of 3–5 points as “not lonely”, whereas individuals with a score of 6–9 points are defined as “lonely” [21], allowing a simplified description of the cohort. 

### 2.3. Independent Variables

To support regression analysis undertaken with the data, several independent variables have been included in the study. Basic demographic factors such as sex (male/female), age (in years), marital status (single; divorced; widowed; married, living separately from spouse), level of education (according to the CASMIN classification: primary, secondary, tertiary education [22]) and country of origin (Germany, neighboring countries, other countries) were analyzed. In addition, we included chronic alcohol consumption as indicated by increased levels of carbohydrate-deficient transferrin (CDT, CDT-levels > 2.5% were defined as elevated [23,24]), duration of homelessness (in months) and the frequency of sharing a sleeping space with more than three people (measured with a four-point Likert scale: 1 = “never” to 4 = “always”). Furthermore, we included the self-perceived risk of contracting COVID-19 (measured on a five-point Likert scale from 1 = “very low” to 5 = “high”) into our analysis. 

### 2.4. Statistical Analysis

First, characteristics of the analyzed sample were described. The UCLA-3 score was displayed as a continuous sum score as well as dichotomously distinguishing between lonely and not lonely. Second, multiple linear regressions were used to identify determinants of UCLA-3 sum scores. To address potential bias caused by missing values, we checked our findings by performing full information maximum likelihood analysis [25]. Statistical analysis was performed using Stata 15.0 (Stata Corp, College Station, TX, USA). Significance was defined as a *p*-value of <0.05 throughout all tests.

## 3. Results

### 3.1. Sample Characteristics

Sample characteristics for our analytical sample are given in Table 1. The sample used for statistical analysis was composed of 130 individuals; 79.2% of the individuals were male, and the mean age was 44.1 years (SD: 13.0; range: [19;86]). The mean time of homelessness was 62.3 months (SD: 120.0 range: [1;720]). Most individuals were born in Germany (52.0%), whereas 25.2% of the individuals emigrated from countries neighboring Germany, and 22.8% of the individuals emigrated from other EU and non-EU countries. A detailed description of all the independent variables included in the regression analysis is shown in Table 1. 

In the analytical sample, mean perceived loneliness indicated by the sum score of the UCLA-3 questionnaire was 4.43 (SD: 1.93). Defined by a sum score of 6 or above, 48.5% of the individuals felt lonely, whereas 51.5% of the individuals were defined as not lonely.

### 3.2. Linear Regression Analysis

Linear regression analysis was performed using the UCLA-3 sum score as an outcome measure. Results are illustrated in Table 2. Regressions revealed that decreased loneliness was associated with female gender (β = −1.07, *p* = 0.01) and being from a neighboring country of origin (β = −1.48, *p* = 0.00). Furthermore, increased loneliness was associated with being single (β =1.33, *p* = 0.00), a high frequency of sharing a sleeping space with more than three people (β = 0.42, *p* = 0.02) and a high self-perceived risk of contracting COVID-19 (β = 0.41, *p* = 0.02). By contrast, there was no association between the feeling of loneliness and age, educational level, chronic alcohol consumption or the frequency of sharing a sleeping space with multiple persons. To address missing data, a full information maximum likelihood approach was used in sensitivity analysis, and detailed results are shown in Appendix A. In terms of significance and effect sizes, the findings did not change substantially. However, it is worth noting that the link between the country of origin and loneliness disappeared.

To gain a more complete picture of the data, linear regressions were also performed using divorced as a reference group for marital status. However, no association between loneliness and marital status was observed within these regressions. Detailed results of the analysis are displayed in Appendix A. 

## 4. Discussion

Our study investigated the feeling of loneliness among homeless people living in Hamburg during the first wave of the COVID-19 pandemic. By using linear regression models, we identified being of female gender and from a neighboring country of origin to be associated with a decreased likelihood of loneliness, whereas being single and having a high frequency of sharing a sleeping space with more than three people, as well as a high self-perceived risk of contracting COVID-19, increased the likelihood of loneliness. Taking such characteristics into consideration provides a way for identifying homeless individuals at risk of feeling lonely who can possibly benefit from targeted interventions. These interventions may include the improvement of social skills, the enhancement of social support, offering opportunities for social interaction and addressing maladaptive social cognition [26].

Multiple commentaries have been published highlighting the challenges in caring for the homeless population and the public health reasons for supporting homeless individuals during the COVID-19 pandemic [11,16,17]. However, scientific data on the psychiatric and somatic health of homeless individuals before and during the pandemic are still scarce, as highly mobile and vulnerable populations can be hard to access for research purposes. Moreover, to our knowledge, no study addressing loneliness among homeless individuals during the COVID-19 pandemic has been published. Of note, at the time of data collection, only about 5100 cases of COVID-19 were reported in a population of 1.8 million people living in Hamburg [27]. Such low numbers of cases were achieved by drastic restrictions [28] which caused shelters, soup kitchens and medical practices offering care to homeless individuals to close [15]. Despite the marked influence of the pandemic on the everyday life of homeless individuals, the self-perceived risk of contracting COVID-19 was low, and the frequency of sharing a sleeping space, possibly reflecting the inability to practice social distancing when living on the streets or in need of an emergency shelter, remained high. This observation further underlines the increased susceptibility of the homeless population during the COVID-19 pandemic. Therefore, highlighting the prevalence of loneliness among homeless individuals during a pandemic is of high importance, as loneliness may prevent homeless individuals from connecting with counseling and social workers [18]; a contact that is crucial for adequate prevention and treatment of SARS-CoV-2 infections. Thus, our study adds valuable knowledge to the scientific community. 

Our data show that nearly half the included homeless individuals experienced loneliness during the COVID-19 pandemic. Interestingly, data acquired from the general adult population of Germany in 2012 only identified 10.5% of the population to be lonely, based on ratings on a single item scale [5]. Because of methodological differences between this study and our data, and because of the plausible influence of the COVID-19 pandemic on loneliness, comparability of these two datasets is limited. Nevertheless, it underlines a discrepancy in the prevalence of loneliness between the general population and the homeless population that is well described in other countries. For example, studies conducted in Canada and Australia also revealed exceptionally high prevalence rates of loneliness among homeless individuals [7,8,9]. Literature proposes that self-perceived social isolation, independent of the objective amount of social interaction, leads to the feeling of loneliness [1]. Homeless individuals often have only weak social networks and a tendency to alienate close contacts [7]. In addition, social isolation among homeless individuals might be further aggravated by mental illnesses, substance abuse, distress and a lack of meaningful activity and motivation, which contribute to difficulties in maintaining relationships [7]. Furthermore, vulnerable and stigmatized groups tend to experience more loneliness, compared with the general population [19]. Hence, self-perceived social isolation and stigmatization of homeless individuals might therefore explain a higher prevalence of loneliness among homeless individuals compared with the general public. Of note, the physical presence of other individuals is not sufficient to protect from loneliness [1]. Interestingly, this finding is also reflected in our data, where we observed a higher likelihood of loneliness in individuals who frequently shared a sleeping space with more than three people. This finding might be influenced by the reality that, with high demand for sleeping spaces during the pandemic, shelters for homeless individuals were mostly unable to consider individual preferences for companionship when distributing unoccupied sleeping spaces.

In 2013, Patanwala and colleagues conducted an observational study including 283 homeless individuals living in California. Using the same version of the UCLA-3 questionnaire, they identified 39.6% of the homeless individuals to be lonely [10]. Interestingly, this prevalence appears lower than the 48.5% of individuals perceiving loneliness we identified in our study. Most certainly, comparability of these data is limited due to differences in the analyzed cohorts. For example, the included individuals show a different demographic structure. In addition, as trials were conducted in different states, cultural differences and deviating social care offers must be taken into account. Nevertheless, it is most plausible that increased loneliness displayed in our data is associated with the COVID-19 pandemic. The hypothesis of increased loneliness during the COVID-19 pandemic is supported by several studies. Data from the German socioeconomic panel revealed increasing loneliness in the general population during the first wave of the COVID-19 pandemic. By yearly application of the UCLA-3 questionnaire from 2015 to 2020, an elevation of loneliness by two standard deviations was detected in April of 2020 [6]. Furthermore, a study conducted in the UK by Killgore and colleagues showed an increase in loneliness from April to September of 2020, associated with restrictions implemented during the pandemic [29]. Supporting the hypothesis that the loneliness observed in our dataset was associated with the COVID-19 pandemic, we found the self-perceived risk of contracting COVID-19 to be a determining factor of increased loneliness in the homeless cohort. This might be explained by stricter practicing of social distancing in those with a high self-perceived risk of contracting COVID-19. However, further studies acquiring longitudinal data are crucial to determine the impact of the ongoing pandemic on loneliness among homeless individuals. 

Strikingly, we identified female gender to be associated with decreased loneliness. This finding is in contrast with data about the general population of Germany, showing a higher prevalence of loneliness in females before [5] and during the COVID-19 pandemic [6]. It is noteworthy to consider that homeless women might have special considerations to take into account. In general, far fewer women than men sleep rough or stay in an emergency shelter, a finding which is also represented in the demographics of our sample. Compared with homeless men, homeless women are more likely to be accompanied by small children and only seldomly sleep on the street [30]. In addition, when women do sleep rough, they tend to do so for much shorter periods and are likely to overcome unsheltered homelessness more rapidly [31]. It is possible that frequently having company, shelter and being homeless for shorter periods of time might contribute to a decreased loneliness in the population of homeless women, compared with homeless men. Counterintuitively, individuals born in a neighboring country displayed lower levels of loneliness compared with individuals born in Germany. However, since this association vanished in sensitivity analysis, it should be interpreted with great caution. Published literature offers no clear explanation on why being from a neighboring country of origin should be associated with decreased loneliness. Qualitative interviews conducted in 2016 indicate that homeless individuals might identify friends, rather than family, to be the source of constant practical and emotional support [32]. It is possible that, as immigration from neighboring countries was frequently reported in our sample (25.2%), these individuals benefit emotionally from forming subgroups with other homeless individuals of a similar cultural background. 

As expected, in comparison with homeless individuals that were married, but living separately from their spouse, single individuals reported higher levels of loneliness. The link between marital status and loneliness is similarly reflected in data acquired from the general public and is mostly dependent on the size of a person’s social network [33]. Therefore, individuals living in singlehood report loneliness more frequently than married individuals, even if their marriage was not perceived as a satisfying relationship [34]. It is of note that within our cohort of homeless individuals, no one reported to be married and living with their spouse. Data from the general population show that individuals who are separated also show a higher prevalence of loneliness compared with those in successful marriages [33]. Thus, we assume that our reference group of homeless individuals that are married, but live separately from their spouse, might per se have an elevated risk of loneliness.

Regarding the methodology of this study, some strengths and limitations are worth noting. When interpreting the data on loneliness with regards to the COVID-19 pandemic, it is important to keep in mind that our data are cross-sectional. We are therefore not able to determine the impact of the COVID-19 pandemic on loneliness among homeless individuals. Nevertheless, highlighting high prevalence of loneliness and identifying determining factors opens opportunities to identify and support homeless individuals at risk of feeling lonely during a worldwide crisis. Of note, other intervening variables such as psychiatric comorbidities alongside chronic alcohol consumption, or the form of sheltering, may impact the feeling of loneliness among homeless individuals. Further studies are needed to clarify these associations. In our study loneliness was measured using the three-item UCLA-3 questionnaire, which is a widely used instrument with high sensitivity and specificity [20]. This instrument was chosen for fast and reliable evaluation of loneliness, providing a questionnaire that was both short and easy to comprehend [11]. More detailed questionnaires, like the 11 or 20 item UCLA loneliness scale, offer further information regarding the feeling of loneliness [35]. Furthermore, as our study used the international version, which includes a 3-point Likert scale, and other studies conducted in Germany used a 4-point Likert scale [6], comparability of our data to data from the general population of Germany is limited. Despite the shortness of the questionnaire, missing values occurred. Therefore, we confirmed our findings from linear regression by using full information maximum likelihood analysis. Our analytical sample consisted of 130 homeless individuals in the sensitivity analysis, a vulnerable group that is hard to reach. Furthermore, we noted a very high response rate of 98%. Looking at the composition of the analytical sample, it is plausible that selection bias occurred, as participants were recruited in and near to open shelters. Thus, it is very likely that individuals with bad health and individuals not utilizing any public, social or health care offers are underrepresented in this study. 

To our knowledge, no study has ever evaluated the feeling of loneliness among homeless individuals during the COVID-19 pandemic. Our data add initial insights in an emerging area of concern for public health professionals, social workers and counsellors.

## 5. Conclusions

This study highlights the magnitude of loneliness among homeless individuals living in Hamburg, Germany, during the first wave of the COVID-19 pandemic. Regression analysis showed increased loneliness to be associated with male gender, being single, being originally from Germany, a high frequency of sharing a sleeping place with more than three people and a high self-perceived risk of contracting COVID-19. Such descriptions of factors determining loneliness may help to identify homeless individuals at risk of feeling lonely, opening up opportunities to identify individuals who could benefit from targeted interventions. 

## Figures and Tables

**Table 1 ijerph-18-03035-t001:** Sample characteristics (*n* = 130).

Independent Variables	*n*/Mean (%/SD)
Gender	
*Male*	103 (79.2%)
*Female*	27 (20.8%)
Age	44.1 (13.0)
Family status	
*Single*	82 (67.8%)
*Divorced*	23 (19.0%)
*Widowed*	8 (6.6%)
*Married, living separately from spouse*	8 (6.6%)
Education (CASMIN classification)	
*Primary education*	10 (8.3%)
*Secondary education*	103 (85.8%)
*Tertiary education*	7 (5.8%)
Country of origin	
*Germany*	64 (52.0%)
*Neighboring country*	31 (25.2%)
*Other country*	28 (22.8%)
Duration of homelessness (months)	62.3 (120.0)
Frequency of sharing a sleeping space with more than 3 people (from 1 = never to 4 = always)	2.9 (1.2)
Alcohol consumption	
*Absence of chronic alcohol consumption (CDT ≤ 2.5)*	79 (64.2%)
*Presence of chronic alcohol consumption (CDT > 2.5)*	44 (35.8%)
Self-perceived risk of contracting COVID-19 (from 1 = very low to 5 = high)	1.8 (1.0)

**Table 2 ijerph-18-03035-t002:** Determinants of loneliness among homeless individuals during the COVID-19 pandemic: Findings of multiple linear regressions.

Independent Variables	Coeff.	SD	T	P > t	95% CI	
Gender: Female (Ref.: male)	−1.07 *	0.42	−2.56	0.01	−1.91	−0.24
Age	0.00	0.02	0.11	0.91	−0.03	0.03
Marital Status (Ref.: married, living separately from spouse):						
*Single*	1.33 **	0.43	3.06	0.00	0.47	2.19
*Widowed*	0.88	1.10	0.80	0.43	−1.31	3.06
*Divorced*	1.05	0.58	1.82	0.07	−0.10	2.20
Level of Education (CASMIN Classification) (Ref.: Primary):						
*Secondary*	−0.99	0.61	−1.62	0.11	−2.20	0.22
*Tertiary*	0.39	0.96	0.41	0.68	−1.51	2.30
Country of origin (Ref.: Germany):						
*Neighboring country*	−1.48 **	0.51	−2.93	0.00	−2.49	−0.48
*Other EU- and non-EU countries*	−0.34	0.57	−0.60	0.55	−1.49	0.80
Duration of homelessness (months)	0.00	0.00	0.83	0.41	−0.00	0.01
Share a sleeping space with more than three persons: (from 1 = never to 4 = always)	0.42 *	0.17	2.45	0.02	0.08	0.76
Chronic alcohol consumption (CDT >2.5%): Presence (Ref.: absence)	0.18	0.53	0.34	0.74	−0.87	1.22
Self-perceived risk of contracting COVID-19 (from 1 = very low to 5 = high)	0.41 *	0.18	2.30	0.02	0.06	0.77
Constant	3.48	0.98	3.54	0.00	1.52	5.43
Observations	94					
R^2^	0.28					

Unstandardized regression coefficients are displayed; robust standard errors in parentheses: ** *p* < 0.01, * *p* < 0.05.

## Data Availability

The datasets analyzed in the current study are not publicly available due to ethical restrictions involving patient data but may be available from the corresponding author upon reasonable request.

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
