# Peer review of "Loneliness among Homeless Individuals during the First Wave of the COVID-19 Pandemic"

_ijerph, 2021, doi:10.3390/ijerph18063035_

Round 1
Reviewer 1 Report
Thank you for the opportunity to re-review your paper. It remains a paper of great interest to the readership of this journal and beyond.
I can see you have taken onboard most of the comments provided in the detailed reviews you received.
The paper, however, STILL needs a thorough proof reading/editing by someone who is not so close to the content, as there remain too many errors of grammar, incorrect wording (from v form; Off not Of etc) and issues with inconsistency. I suggest one of the more senior or less involved authors take on this role, or you ask a trusted colleague to do this. Otherwise, it might be worth paying a proofer to do the job; someone with an eye for detail and good grasp of English.
There remain a couple of areas in the paper where the message being conveyed isnt quite clear or is even contradictory to me. Please attend to these. I have marked them up again in comments on the PDF.
Again, thank you for submitting this study. I hope after the next round of changes are made attentively, it makes it through to publication.

Author Response
We thank Reviewer 1 for the thoughtful and highly valuable comments that helped to further improve our work. Here, we provide a point-by-point response:
Thank you for the opportunity to re-review your paper. It remains a paper of great interest to the readership of this journal and beyond. I can see you have taken onboard most of the comments provided in the detailed reviews you received.
We thank the Reviewer for the positive feedback and appreciate the effort made to re-review our work.
The paper, however, STILL needs a thorough proof reading/editing by someone who is not so close to the content, as there remain too many errors of grammar, incorrect wording (from v form; Off not Of etc) and issues with inconsistency. I suggest one of the more senior or less involved authors take on this role, or you ask a trusted colleague to do this. Otherwise, it might be worth paying a proofer to do the job; someone with an eye for detail and good grasp of English.
We agree with the evaluation of Reviewer and thank them for the detailed corrections in the PDF. To further improve the quality of our text we asked a native English-speaking colleague to proofread the manuscript and therefore extended the acknowledgement section (line 365-366).
“Furthermore, we thank Mrs. Morsal Sabihi for carefully proofreading the manuscript.”
We hope the changes made throughout the document will significantly improve the consistency and clarity.
There remain a couple of areas in the paper where the message being conveyed isnt quite clear or is even contradictory to me. Please attend to these. I have marked them up again in comments on the PDF.
Once again, we thank the Reviewer very much for the time and effort spent on these incredibly helpful, detailed corrections. All comments in PDF could be fully addressed and are highlighted in green throughout the manuscript (please see attachment). As requested by the journal, the supplementary tables were uploaded in a sperate file in the supplementary section. According to the suggestions Reviewer 1 made regarding the tables within the manuscript, we also revised our supplementary tables.
Again, thank you for submitting this study. I hope after the next round of changes are made attentively, it makes it through to publication.
We thank Reviewer 1 for the helpful comments, the kind words, and the encouragement.

Reviewer 2 Report
Introduction
- Loneliness is not a feeling, but rather an experience with cognitive, emotional and behavioral manifestation. Your description of that experience needs to be included.
- The whole section about loneliness [first paragraph in Introduction] is in my opinion too brief.
- The definition, and description of homeless people, is very brief and does not actually describe what homelessness is.
Discussion
- It seems unreasonable, to me, to compare the number of lonely homeless in 2020 to the general population eight years (2012) earlier, under very different conditions.
- You stated that “We are therefore not able to determine the impact of the COVID-19 pandemic on loneliness among homeless individuals“. So it negs the question what is the purpose of the study, beyond indicating the higher rate of loneliness, compared to the general population.
- You, also, did not list some limitations of this study, like other, intervening, variables that may affect the results such as psychiatric problems, whether the homeless sleep somewhere or actually on the street, their marital situation, etc.
Author Response
Reviewer 2
We thank Reviewer 2 for the thoughtful and highly valuable comments that helped to further improve our work. Here, we provide a point-by-point response:
Loneliness is not a feeling, but rather an experience with cognitive, emotional and behavioral manifestation. Your description of that experience needs to be included. The whole section about loneliness [first paragraph in Introduction] is in my opinion too brief.
We thank Reviewer 2 for this valuable comment and further strengthened our paragraph about loneliness in line 29-40. Following the advice of Reviewer 2 we included a more detailed description of the cognitive, emotional, and behavioural manifestations of loneliness:
The feeling of loneliness describes a discrepancy between actual and desired social relationships and is considered a major public health problem [1,2]. Loneliness is associated with multiple somatic and psychiatric illnesses, as well as premature death [3]. Meta-analysis found elevated relative risk for hypertension, cardiovascular disease and post myocardial infarction mortality in people experiencing loneliness. Furthermore, loneliness is associated with reduced well-being an higher risk for depression, suicide and dementia is associated with personality disorders, psychosis, depression, and suicide [4]. The feeling of loneliness also impairs cognitive performance, increases the risk of cognitive decline, and reduces executive control. Further highlighting the devastating consequences of loneliness, experimental evidence revealed that the feeling of loneliness also increases perceived stress, fear of negative evaluation, anxiety, and anger by reducing optimism and a person’s self-esteem [1].
The definition, and description of homeless people, is very brief and does not actually describe what homelessness is.
We agree with Reviewer 2 and consequently elaborated the description of homeless individuals, by describing reasons for homelessness and specifying common health challenges of homeless individuals (compare line 51-55).
According to the federal state law of the City of Hamburg, a homeless individual is defined as someone who does not have stable housing and has been sleeping rough, or in emergency shelters for most days within the last month. Individual reasons for homelessness are diverse, but frequently include poverty, family problem, substance misuse and the availability of low-cost housing [11]. Compared with the general population, homeless individuals have higher rates of premature death, increased prevalence of infectious, and chronic diseases as well as high prevalence of mental disorders [11]. In 2018, an estimated 41000 homeless individuals were living in Germany [12], thereof 6500 in the federal state of Hamburg, the second largest city in Germany [13]
It seems unreasonable, to me, to compare the number of lonely homeless in 2020 to the general population eight years (2012) earlier, under very different conditions.
We thank Reviewer 2 for this valuable comment and agree that comparison between the general population before and the homeless population during the COVID-19 pandemic are rather difficult and should be interpreted with great caution. We therefore further stressed the limitations of this comparison in line 215-223
Interestingly, data acquired from the general adult population of Germany in 2012 only identified 10.5% of the population to be lonely based on ratings on a single item scale [5]. Because of methodological differences between this study and our data, and because of the plausible influence of the COVID-19 pandemic on loneliness, the comparability of these two datasets is limited. Nevertheless, it underlines a This discrepancy in the prevalence of loneliness between the general population and homeless population that is well described in other countries seems to have existed independently of the COVID-19 pandemic, as past For example, studies conducted in Canada and Australia also describe revealed exceptionally high prevalence rates of loneliness among homeless individuals [7-9]
Furthermore, we thank Reviewer 2 for the attentive comment on the age of the cited study. More recent data on loneliness in the German general population is published yearly by the German Socio-Economic Panel (SOEP). Unfortunately, in the SOEP uses a different rating scale for the UCLA-3 questions (4- point Likert scale instead of the international 3- point Likert scale) that does not allow discrimination between “lonely” and “not lonely”. We therefore chose the study from Beutel, et al despite its age.
You stated that “We are therefore not able to determine the impact of the COVID-19 pandemic on loneliness among homeless individuals“. So it negs the question what is the purpose of the study, beyond indicating the higher rate of loneliness, compared to the general population.
We agree with Reviewer 2 that it is of great interest to determine the impact of COVID-19 pandemic on loneliness among homeless individuals. Unfortunately answering this question would require longitudinal data and is therefore beyond the scope of our project. We therefore decided to highlight the relevance of our cross-sectional data to the scientific community by modifying line 191-196 and 329-332:
While multiple Multiple commentaries have been published highlighting the importance challenges in caring for the homeless population and the public health reasons for supporting homeless individuals during the COVID-19 pandemic [17,11]. However, scientific data on psychiatric and somatic health of homeless individuals before and during the pandemic is still scarce, as highly mobile, and vulnerable populations can be hard to access for research purpose. Moreover, to our knowledge, no study addressing loneliness among homeless individuals during the COVID-19 pandemic has been published.
Nevertheless, as To our knowledge, no study has ever evaluated the feeling of loneliness among homeless individuals during the COVID-19 pandemic. Our data adds initial insides insights in an emerging area of concern for public health professionals, social workers, and counsellors.
You, also, did not list some limitations of this study, like other, intervening, variables that may affect the results such as psychiatric problems, whether the homeless sleep somewhere or actually on the street, their marital situation, etc.
We thank the Reviewer for this highly valuable comment and further specified the limitations of our study regarding intervening variables in line 308-310.
Of note, other intervening variables such as psychiatric comorbidities besides alongside chronic alcohol consumption or the form of sheltering may impact the feeling of loneliness among homeless individuals. Further studies are needed to clarify these associations.
Round 2
Reviewer 2 Report
Revisions executed as required.
This manuscript is a resubmission of an earlier submission. The following is a list of the peer review reports and author responses from that submission.
Round 1
Reviewer 1 Report
The research in very original and presents an interesting matter of investigation. The manuscript is well written.
My only comment is that they mention in the method taking a nasopharyngeal swap, and a physical examination. Then, they do not present results regarding those procedures.
I suggest reporting some results, otherwise, take out from the methods. It will be interesting to know the prevalence of Covid-19 if the nasopharyngeal swab aimed to do this diagnosis.
Reviewer 2 Report
Thank you for writing a short paper on a really interesting topic. I have enjoyed reading it and offer a range of comments on the content to bring it up to the standard for the journal it has been submitted to. I encourage you to make these changes, as it is a useful and timely contribution and I believe we should support students in particular to contribute as authors where we can.
The paper does need some work though, as you will see from my detailed comments in the PDF attached. I hope these make sense to you and you can see them as critiques designed to improve readability, flow, argument, appeal.
In addition to the comments on the paper, I offer the following thoughts:
It might also be helpful to add a paragraph to the paper on limits to the study, reinforcing the point that this is a paper that provides some initial insights in an emerging area of concern (in a changing world!).
I also note that social exclusion and social isolation are among the key words, but pretty sure they arent discussed as concepts or even used as terms. Perhaps change the keywords or add some commentary around them.
There is also a literature on the life impacts of loneliness you could refer to here... equating it with the impacts or smoking or obesity. That might help with framing this better/significance etc.
Once you have reworked the paper in line with the comments, I would be pleased to reread it. And please re-look at the abstract and introduction when you have reworked the main paper.

Reviewer 3 Report
Bertram F et al.: Loneliness among homeless individuals during the COVID-19 2 pandemic: a cross-sectional monocentric study.
This is a cross-sectional survey of loneliness among homeless individuals in Hamburg during COVID pandemic. Finally data from 130 participants were analyzed. A response rate of 98% was reached. Beside demographic variables country of origin, alcohol consumption, duration of homelessness, frequency of sharing a sleeping place and perceived risk of contracting COVID were the independent variables in the analysis.
While I agree with authors conclusion that lonely homeless people deserves higher attention during COVID pandemic, I have some concerns regarding the methodology of this study.
Ninety-eight percent response rate in a survey like this is surprisingly high. Did participants receive any kind of compensation for entering the study?
While the study was conducted during the COVID pandemic, I am not convinced that the detected level of loneliness is really connected with the pandemic. For confirming this correlation authors should organize a follow up survey and compare the level of loneliness before and during the pandemic. Or alternatively authors should demonstrate how the way of living of the participants was changes as a result of the pandemic. Sharing a sleeping place with more than 3 people was frequent in the sample indicating that social distancing was not practiced in this subgroup. Perceived risk of COVID contracting COVID was also relatively low. The results that those who were single and who perceived higher risk of contracting COVID (and probably kept stricter social distancing) were lonelier are predictable.
Frequency of different psychiatric disorders among homeless individuals is rather high and can result loneliness as well. Unfortunately only alcohol consumption was studied from psychiatric morbidity, while a number of other disorders may also have impact on the social network and the feeling of loneliness of homeless people.